# Application of Exploratory Factor Analysis and Item Response Theory to Validate NHANES ADL Scale in Patients Reporting Rheumatoid Arthritis

**DOI:** 10.3390/pharmacy10060138

**Published:** 2022-10-22

**Authors:** Prashant Sakharkar, Najma Ansari

**Affiliations:** 1College of Science, Health and Pharmacy, Roosevelt University, Schaumburg, IL 60173, USA; 2Karmonas Cancer Institute, Detroit, MI 48201, USA

**Keywords:** rheumatoid arthritis, activities of daily living, ADL, item response theory, IRT, NHANES, National Health and Nutritional Examination Survey, scale reliability, survey validation

## Abstract

**Background**: Several instruments are used for measuring functional limitations among rheumatoid arthritis (RA) patients. However, these instruments are incongruously assessed for their psychometric properties. The National Health and Nutritional Examination Survey (NHANES) uses a generic questionnaire to assess the activities of daily living (ADL) to measure functional limitations among its participants. The psychometric properties of the NHANES-ADL scale were evaluated using a patient examination and survey data. **Methods**: NHANES-ADL scale was assessed for its internal consistency and factor structure. Scale reliability was assessed with Cronbach’s alpha reliability coefficient. Principal component analysis with Promax rotation was used to obtain factor structure. Confirmatory factory analysis was used to calculate fit indices. The graded item response theory model was used to estimate item discrimination, difficulty, and test information. **Results**: Our sample included 1132 individuals with RA. Exploratory factor analyses of 19-item NHANES ADL scale produced one factor solution and accounted for 35% of variance. The Cronbach alpha of this scale was 0.92. The results of graded item response model indicated items performing well discriminating high and low level of functional ability. A higher slope (α) reflected stronger ability of items to discriminate across the continuum. **Conclusions**: The NHANES ADL scale showed good reliability, single dimensionality, and validity in RA patients. Studies should explore its test-retest reliability and its ability to reliably measure functional change over time in the future.

## 1. Introduction

Approximately 1% of the world population and about 1.5 million people in the United States suffers from rheumatoid arthritis (RA) [1]. For more than 15 years, it is the most frequently reported main cause of disability among U.S. adults [2]. According to the 2016–2018 National Health Interview Survey (NHIS) data, an estimated 58.5 million (23.7%) of 18 years and older adults reported arthritis, whereas 25.7 million reported arthritis attributable activity limitations [2]. The prevalence of activity limitations attributable to RA is projected to increase 49% from 2012 to 2040 [3].

RA is an inflammatory chronic progressive disease. Despite the progressive nature of RA, patients experience high and low levels of functional ability. Chronic pain and functional limitations are the most severe effects of RA [4,5]. RA patients often perceive a reduced quality of life. This can be attributed to personal beliefs, loss of a level of independence, and deteriorating physical health compared to the population that is healthy [6]. The restriction of daily normal activities impacts RA patients’ mood and social activities, leading to physical and emotional challenges. RA patients have shown significantly greater physical limitations compared to heathy people [6,7].

Inability to perform usual activities was reported by nearly 43% of 50 million adults that are diagnosed with arthritis. Similarly, 31% of working-age adults report similar limitations. About 40% of adults with RA found it very difficult or impossible to do at least one of the following nine activities of daily living (ADL): stooping, bending, or kneeling; standing more than 2 h; walking a quarter of a mile; pushing a heavy object; climbing a flight of stairs; lifting or carrying 10 pounds; siting for more than 2 h; reaching above your head and grasping small objects [8].

It is evident that, future health outcomes and treatment response in RA patients can be best predicted by subjective patient and physician measures. Functional status impacted with bone and joint conditions is commonly assessed using various health assessment instruments or questionnaires. These questionnaires or instruments assess ADL including self-care, mobility, housework, and transportation, etc. The Short Form-36 (SF-36) has been a widely used, multidimensional non-disease-specific ADL instrument. The Health Assessment Questionnaire Disability Index (HAQ) is considered to be the gold standard for measuring functional activity in RA patients [9]. However, its large number of items (41) and relatively complicated scoring method make its use difficult in clinical practice.

There are several generic and disease specific instruments used in measuring functional status among RA patients, such as the Barthel index for activities of daily living, the Groningen activity restriction scale (GARTH), the World Health Organization Disability Schedule-II (WHODAS-II), the Health Assessment Questionnaire Disability Index (HAQ-DI) Health Assessment Questionnaire II (HAQ-II), the shortened arthritis impact measurement scales (AIMS), the Multidimensional Health Assessment Questionnaire (MDHAQ-10-ADL), MDHAQ-14-ADL, and the Sickness Impact Profile (SIP-RA) to name few [9,10,11,12,13,14,15,16,17,18,19]. Some of these tools are inconsistently assessed for their psychometric properties. The majority were assessed for reliability, some were assessed for their factor structure, test-retest reliability and responsiveness, and a few were assessed for item discrimination, difficulty, and sensitivity [16,17,18,19].

The National Health and Nutritional Examination Survey (NHANES) uses a generic questionnaire to assess ADL. Its psychometric properties were assessed in patients with cervical pain, low back pain, severe headache, stroke, and general population previously [20,21,22,23,24]. These analyses revealed challenges of discriminatory properties and sensitivity associated with certain ADL questionnaire items. Despite frequent use of NHANES-ADL scale, the psychometric properties of items used for the assessment of functional limitation within this dataset remain unexplored. We performed a validation study to assess the psychometric properties of the NHANES-ADL scale in RA patients, using a pre-existing patient examination and survey data.

## 2. Materials and Methods

### 2.1. Study Population

The NHANES assesses the health and nutritional status of children and adults in the United States. A cross-section of nationally representative non-institutionalized US civilian population is surveyed on health and nutrition [25]. The survey combines interviews and physical examination components. The interview portion includes collection of demographics data, data on socio-economic status, health, and data related to nutrition [26]. The physical exam components include medical, dental, physiological, and laboratory tests, conducted by trained medical personnel performed in a mobile examination center [27]. During the physical examination, participants were interviewed about chronic conditions including RA. We used the NHANES data from 2011–2018 in this study. Approximately 5000 to 10,000 non-institutionalized individuals from 15 counties across the US sampled each two-year cycle. Individuals who participated in three cycles of NHANES survey was considered an adequate representative sample for our study [28]. Similarly, sample sizes exceeding 500 is considered to achieve adequate precision for certain graded response models in simulation-based studies [29].

Out of 10,714 participants who reported having arthritis, 2032 of whom reported having RA, a total of 1132 participants with RA who completed the ADL questionnaire and responded to all of the items were included in our final analyses. Participants < 18 years of age and with missing responses to ADL items were excluded.

### 2.2. ADL Assessment

The NHANES survey on functional abilities includes questions related to performing ADLs [14]. This questionnaire is designed by NHANES researchers to measure functional status reflecting the unidimensional latent constructs of ADL. Of the 20 items included in the ADL questionnaire, 19 items were selected for our analysis (Appendix B). The item: “how much difficulty do you have managing money”, was not included. Responses to these items were graded on a 4-point Likert-type scale with a score ranging from one to four representing the level of difficulty in performing various tasks (1 = no difficulty, 2 = some difficulty, 3 = much difficulty, and 4 = unable to do) [30]. All individual item scores were added together to get the composite score, with a higher score indicating a greater level of ADL difficulty.

### 2.3. Statistical Analysis

We analyzed the NHANES-ADL scale for its internal consistency, factor structure, and item characteristics with the graded item response theory (G-IRT) model. Statistical analyses were performed using the SPSS 27 (IBM, Chicago, IL, USA) and the STATA 14 (STATA Corp., College Station, TX, USA). Data were reported as means, frequencies, and percentages. Scale reliability was assessed by the Cronbach’s alpha reliability coefficient and its value of 0.7 was considered acceptable [29]. Maximum likelihood estimation was used to determine the dimensionality. Factor structure was assessed using principal factor analysis with Promax rotation. Confirmatory factor analysis was used to calculate fit indices. G-IRT model was used to estimate item discrimination, difficulty, and test information [31,32]. Similarly, the G-IRT model was also used to analyze differences, the item’s ability to discriminate across ADL difficulty levels, including slope parameters, threshold levels, and item characteristic curves. A stronger ability to discriminate across the continuum is indicated by higher slope values (α) [31]. The location of the latent construct was determined by item difficulty or item parameter estimates (β) and identities where each response is most sensitive by indicating the likelihood of endorsing each of the four Likert-types responses [31,32]. The absolute fit indices included the Chi-Squared test, the Root Mean Square Error of Approximation (RMSEA), and the Standardized Root Mean Square Residual (SRMR). The incremental fit indices included comparative fit index (CFI) and Tucker-Lewis Index (TLI). A cut off value of 0.80 was preferred. These indices were found to be the most insensitive to the sample size, model misspecification, and parameter estimates [33,34].

### 2.4. Item Characteristics Curves (ICC)

ICCs were also plotted, where each figure represented a single item, whereas the curves represented four Likert-type responses to the individual items. The x-axis represented the latent constructs of ADL, with a z-score of −3 to +3. A negative score on the x-axis translated to high level of ADL ability. In contrast, an increasingly positive or higher score represented a lower ADL ability. Each of the ICC response had four curves, the first one indicates “no difficulty”, the second “some difficulty”, the third “much difficulty”, and the fourth “unable to do”. Item difficulty, also known as horizontal threshold parameter estimate (β), indicate the likelihood of each Likert-type response between each item. The positive correlation of the individual responses with the latent construct of ADL difficulty was indicated by any value above zero [35].

## 3. Results

Our study sample included 1132 individuals with RA. The respondents mean age was 58.2 ± 0.6 years and 664 (58.7%) were female. Most respondents were non-Hispanic white (35.1%), had a college/associate degree (31.3%), were married (47%) and had an annual family income of <$25,000 (44.2%). Additionally, 89.3% of the respondents reported having some form of health insurance (Table 1).

A 19-item NHANES ADL scale produced one factor solution accounting for 35% variance on exploratory factor analysis using Promax rotation. This suggests most of the items were related to the latent construct of ADL. The largest single factor loading (difficulty in moving large objects, difficulty in standing for a long period, difficulty going to movies/events) strongly suggests the relationship between RA and the disruption of physical and social activity. The Cronbach alpha of 0.92 of the single dimensional scale was well above the accepted value of 0.70 (Table 2).

The results of the item response theory, using the graded response model, indicated that all items discriminated well between respondents with a high and low level of functional ability in performing ADLs. The majority of items showed stronger discrimination, which included: difficulty doing chores around the house, preparing meals, walking from one room to the other, standing up from an armless chair, getting in and out of bed, difficulty using silverware or holding a cup, dressing up or doing buttons, standing for extended intervals of time, going out to attend events or participating in social activities, leisure activities, and difficulty moving large objects (Table 3).

All the items and Likert-type responses were located above zero, indicating high levels of ADL limitations and an increased level of sensitivity for the ADL’s latent construct. Our analysis showed the majority of the NHANES-ADL questionnaire items exhibited a stronger performance to discriminate across the scale as indicated by higher slope, α. The ICC of items 9, 10, 13, 15, and 18, had the highest level of endorsement for the response “much difficulty”. In all other curves, except for items 2, 13, and 14, the highest level of endorsement was demonstrated by “some difficulty”. This further suggested the highest probability of being associated with ADL limitation. The ICCs for items 2, 13, and 14, were less discriminating as reflected in the relative flattening of item curves. However, the values of slope were less convincing, but still acceptable (Figure 1).

The NHANES-ADL questionnaire revealed singular dimensionality and strong internal reliability at scale level (Appendix A). Using item analysis, all NHANES-ADL questionnaire items were discriminatory of ADL limitations. Scale statistics included a mean score of 21.4 ± 5.5 with a variance of 31.3. (Appendix A). The SRMR, CFI and TLI of 0.0510, 0.890 and 0.874, respectively, indicated a good fit; whereas the RMSEA of 0.0821 (95%CI: 0.0788, 0.0855) indicated a fair fit. (Appendix A).

## 4. Discussion

Patients are considered the best and most accurate source for quantifiable data related to functional limitations, pain, fatigue, and mental health etc. Functional limitation is the most significant prognostic clinical measure of long-term outcomes in RA patients measured by self-reported questionnaire. The NHANES uses a generic questionnaire to measure the functional limitation (ADL) among its participants [30]. High and low levels of functional limitations in patients with RA were well discriminated by our IRT analyses. This analyses further explained a wide range of patients’ ability in performing ADLs. In contrast to the classical test theory analysis, IRT is population independent. Furthermore, IRT is less impacted by the condition’s severity among different population characteristics [34]. The item properties do not depend on a representative sample. It assumes the probability that a response chosen is a function of the respondent’s illness, disability, or underlying trait. Moreover, IRT also transforms a single latent trait measure into reliable variables when used to compare various scales. This property thus makes each item transferable from one instrument to another [31]. Our results are similar with the findings of earlier studies where most of the NHANES-ADL scale demonstrated strong discrimination for items 4, 5, 7, 8, 9, 16, and 17, whereas it had an acceptable level of discrimination for items 13, 14, and 18. Nearly all Likert type, except two items, item 2 and 12, showed lower sensitivity in these studies [20,21]. A study by Terman and colleagues showed similar discrimination for items among the NHANES participants, regardless of their disease conditions. Items 5, 7, 12, 16, and 19, exhibited stronger discrimination [24].

Most of the items in our study showed stronger discrimination, which included item 5 (difficulty doing chores around the house), item 6 (preparing meals), item 7 (walking from one room to the other), item 8 (standing up from an armless chair), item 9 (getting in and out of bed), item 10 (difficulty using silverware or holding a cup), item 11 (dressing up or doing buttons), item 12 (standing for extended intervals of time), item 16 (going out to attend events or participating in social activities, leisure activities), and item 19 (difficulty moving large objects). All the items and Likert-type responses indicated high levels of ADL limitations and an increased level of sensitivity for the ADL’s latent construct.

Similarly, most items were sensitive to a mild to moderate ADL limitation level. All of the parameter estimates were to the right of z score “0”. Our results were also consistent with finding that items such as difficulty in stooping, crouching, kneeling, standing for long periods, and moving large objects, had a parameter estimate value below zero, indicating that these items were less sensitive to ADL limitation for lower Likert selection [20,21].

The Likert scale of the NHANES-ADL questionnaire provided greater information distinguishing ADL difficulty level compared to when these responses were transformed into binary and tested using a graded response model in other study [24]. Our study also found self-reported items distinguished participants with an average to above average level of difficulty. This is consistent with the finding of a study by Freedman et al. [36].

IRT have been employed to assess item discrimination, test information, and the sensitivity of questionnaires used in national datasets, including the National Health and Aging Trends Study (NHATS) and the Health and Retirement Study [37,38]. The graded response model used in our study showed better discrimination for each item for better model fit compared to Rasch modeling, which by definition assumes that all discrimination parameters are constrained to one. This can be considered as a major strength of our study. The NHANES-ADL scale can be used to assess physical functioning in practice because of its simplicity and ease of scoring [20,21,22,23,24]. Most of the items in the NHANES-ADL scale showed strong discriminative properties across the entire spectrum of ADL severity.

Our study has several limitations. This was a cross-sectional study, therefore, participants measured functional ability as a snapshot of their ADL at one point in time. Thus, the longitudinal impact of RA on ADL could not be examined. Furthermore, test-retest reliability and the responsiveness of the questionnaire items was not assessed. Additionally, the results cannot be generalized to the individuals with greater levels of functional limitations, or who are institutionalized. Disabled individuals living in institutional settings were excluded in this study. This may have limited the normality of the distribution of the ADL score. Other methodological biases such as non-response, question order, or misclassification, may have underestimated the severity of ADL. In addition, the severity of ADL was self-reported and was not confirmed clinically. The presence of other co-morbidities among participants may have also influenced our results. Individuals with RA accounted for only 2.9% of the total participants, despite the large number of individuals who participated in the NHANES survey. Despite these limitations, our findings suggest a few modifications in this scale may help improve the clinician’s ability to examine and measure functional status more efficiently.

## 5. Conclusions

The NHANES ADL scale showed good reliability, single dimensionality, and validity in RA patients. Strong sensitivity and discrimination across the entire spectrum of functional ability was observed with most of the scale items. The NHANES-ADL scale seems to be a useful tool in measuring ADL among RA patients. Studies should explore its test-retest reliability and its ability to reliably measure clinical change over time in the future.

## Figures and Tables

**Figure 1 pharmacy-10-00138-f001:**
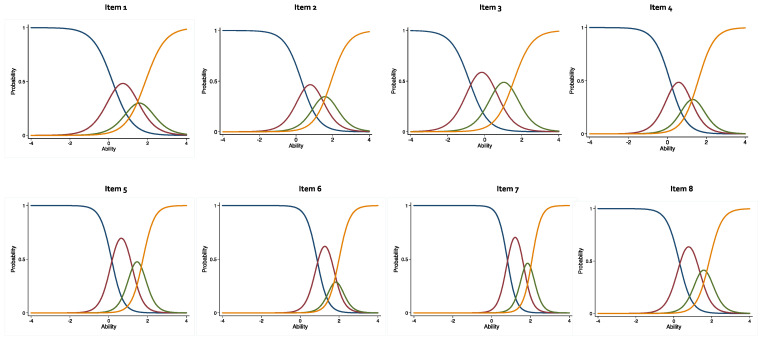
Item characteristic curves for NHANES-ADL scale. Curves represents the probability at varying levels of functional ability among RA participants. The y-axis represents the probability of selection. The x-axis indicates response discrimination toward ADL dysfunction construct.

**Table 1 pharmacy-10-00138-t001:** Demographic characteristics of the NHANES respondents reporting RA (*N* = 1132).

Characteristic	N (%)
Age (years) Mean SD	58.2 ± 0.6
Gender	
Male	468 (41.3%)
Female	664 (58.7%)
Race/Ethnicity	
Mexican American	155 (13.7%)
Other Hispanic	126 (11.1%)
Non-Hispanic White	397 (35.1%)
Non-Hispanic Black	346 (30.6%)
Other Race-Multiracial	108 (9.5%)
Marital Status	
Married	532 (47.0%)
Unmarried	496 (43.8%)
Other	102 (9.0%)
Education Level	
<12th Grade	353 (31.2%)
High School Graduate/GED	273 (24.1%)
Some College/AA Degree	354 (31.3%)
College Graduate	149 (13.2%)
Annual Household Income	
<$25,000	500 (44.2%)
$25,000–$49,999	289 (25.5%)
$50,000–$74,999	91 (8.0%)
$75,000–$99,000	62.0 (5.5%)
≥$100,000	104 (9.2%)
Health Insurance Coverage	
Yes	1011 (89.3%)
No	118 (10.4%)

GED: General Education Development Test; AA: Associate in Arts.

**Table 2 pharmacy-10-00138-t002:** Exploratory factor analysis with Promax rotation and factor loadings of the NHANES ADL scale.

Item	Factor Loadings
Difficulty walking for a quarter mile	0.536
Difficulty walking up ten stairs	0.524
Difficulty stooping, crouching, kneeling	0.592
Difficulty lifting or carrying	0.616
Difficulty doing house chores	0.646
Difficulty preparing meals	0.501
Difficulty walking between rooms	0.407
Difficulty standing up from armless chair	0.600
Difficulty getting in and out of bed	0.635
Difficulty using fork, knife, cup	0.337
Difficulty dressing yourself	0.510
Difficulty standing for long periods	0.661
Difficulty sitting for long periods	0.634
Difficulty reaching up	0.491
Difficulty grasp/holding small objects	0.411
Difficulty going out to movies/events	0.697
Difficulty attending social event	0.626
Difficulty with home leisure activities	0.387
Difficulty moving large objects	0.726

**Table 3 pharmacy-10-00138-t003:** Estimated item parameters for the graded response model and the observed minus expected proportion of responses in each category.

Item	Discrimination	Difficulty			*p*-Value
	α (SE)	β ^1^ (SE)	β ^2^ (SE)	β ^3^ (SE)	
Difficulty walking for a quarter mile	1.97 (0.20)	0.19 (0.09)	1.26 (0.14)	1.89 (0.22)	0.000
Difficulty walking up ten stairs	2.14 (0.23)	0.30 (0.10)	1.24 (0.15)	1.92 (0.32)	0.000
Difficulty stooping, crouching, kneeling	2.02 (0.12)	−0.83 (0.09)	0.51 (0.06)	1.56 (0.11)	0.000
Difficulty lifting or carrying	2.39 (0.21)	0.13 (0.08)	1.02 (0.08)	1.58 (0.10)	0.000
Difficulty doing house chores	3.41 (0.24)	0.14 (0.08)	1.15 (0.08)	1.76 (0.11)	0.000
Difficulty preparing meals	3.59 (0.34)	0.86 (0.08)	1.67 (0.10)	2.00 (0.12)	0.000
Difficulty walking between rooms	4.31 (0.51)	0.81 (0.06)	1.62 (0.09)	2.08 (0.14)	0.000
Difficulty standing up from armless chair	3.00 (0.23)	0.28 (0.05)	1.28 (0.08)	1.87 (0.12)	0.000
Difficulty getting in and out of bed	2.72 (0.28)	0.46 (0.06)	1.55 (0.09)	2.75 (0.23)	0.000
Difficulty using fork, knife, cup	2.06 (0.24)	1.47 (0.15)	2.71 (0.25)	4.46 (0.62)	0.000
Difficulty dressing yourself	2.26 (0.24)	0.62 (0.06)	1.94 (0.18)	2.87 (0.22)	0.000
Difficulty standing for long periods	2.49 (0.21)	−0.51 (0.08)	0.41 (0.06)	1.21 (0.09)	0.000
Difficulty sitting for long periods	1.52 (0.20)	0.27 (0.08)	1.34 (0.16)	2.66 (0.34)	0.000
Difficulty reaching up	1.53 (0.16)	0.35 (0.10)	1.57 (0.16)	2.64 (0.23)	0.000
Difficulty grasp/holding small objects	1.15 (0.14)	0.46 (0.12)	2.42 (0.29)	4.60 (0.62)	0.000
Difficulty going out to movies/events	3.47 (0.27)	0.26 (0.07)	1.17 (0.08)	1.88 (0.11)	0.000
Difficulty attending social event	2.86 (0.30)	0.48 (0.06)	1.33 (0.08)	1.88 (0.12)	0.000
Difficulty with home leisure activities	1.85 (0.23)	1.46 (0.14)	2.75 (0.27)	4.24 (0.61)	0.000
Difficulty moving large objects	2.71 (0.25)	−0.35 (0.08)	0.55 (0.07)	1.10 (0.09)	0.000

α = Alpha refers to an item’s ability to discriminate between different latent levels of ADL (ability). β = Beta represents responses on the 4-point Likert-type scale: ^1^ (from “no difficulty” to “some difficulty”), ^2^ (from “some difficulty” to “much difficulty”), and ^3^ (from “much difficultly” to “unable to do”).

## Data Availability

NHANES data is available in a publicly accessible repository through the Centers for Disease Control and Prevention (CDC) and the National Center for Health Statistics (NCHS). The data presented in this study are openly available in the National Health and Nutrition Examination Survey (NHANES) website at https://wwwn.cdc.gov/nchs/nhanes/ (accessed on 30 August 2021).

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
