# Peer review of "Application of Exploratory Factor Analysis and Item Response Theory to Validate NHANES ADL Scale in Patients Reporting Rheumatoid Arthritis"

_pharmacy, 2022, doi:10.3390/pharmacy10060138_

Round 1

Reviewer 1 Report

Methodological Biases exist

(The Authors must see my remarks)

Author Response

Dear Reviewer, 

We would like to thank you for your valuable suggestions and comments. It has certainly helped us to improve our manuscript. We hope that our revision satisfies your concerns. Please refer to the attachment for our response. 

Thanks again.  

Reviewer 2 Report

 The NHANES-ADL Questionnaire should compared to RA specific parameter such as DSA-28,  the rheumatoid arthritis articular damage score or RAID.

No test-retest reliability available.

Author Response

(The authors gave the same response as above.)

Round 2

Reviewer 2 Report

The manuscript improved much after revision. I have no more comment.